# Validation of the Japanese Version of the Yale Food Addiction Scale 2.0 (J-YFAS 2.0)

**DOI:** 10.3390/nu11030687

**Published:** 2019-03-22

**Authors:** May Thet Khine, Atsuhiko Ota, Ashley N. Gearhardt, Akiko Fujisawa, Mamiko Morita, Atsuko Minagawa, Yuanying Li, Hisao Naito, Hiroshi Yatsuya

**Affiliations:** 1Department of Public Health, Fujita Health University School of Medicine, 1-98 Dengakugakubo, Kutsukake-cho, Toyoake, Aichi 470-1192, Japan; maythet7@gmail.com (M.T.K.); akifuji@fujita-hu.ac.jp (A.F.); liyy@fujita-hu.ac.jp (Y.L.); naitoh@fujita-hu.ac.jp (H.N.); yatsuya@fujita-hu.ac.jp (H.Y.); 2Department of Psychology, University of Michigan, 2268 East Hall, 530 Church Street, Ann Arbor, MI 48109, USA; agearhar@umich.edu; 3Faculty of Nursing, Fujita Health University School of Health Sciences, 1-98 Dengakugakubo, Kutsukake-cho, Toyoake, Aichi 470-1192, Japan; mamorita@fujita-hu.ac.jp (M.M.); mina@fujita-hu.ac.jp (A.M.)

**Keywords:** food addiction, Japan, validation, Yale Food Addiction Scale 2.0

## Abstract

The Yale Food Addiction Scale 2.0 (YFAS 2.0) is used for assessing food addiction (FA). Our study aimed at validating its Japanese version (J-YFAS 2.0). The subjects included 731 undergraduate students. Confirmatory factor analysis indicated the root-mean-square error of approximation, comparative fit index, Tucker–Lewis index, and standardized root-mean-square residual were 0.065, 0.904, 0.880, and 0.048, respectively, for a one-factor structure model. Kuder–Richardson α was 0.78. Prevalence of the J-YFAS 2.0-diagnosed mild, moderate, and severe FA was 1.1%, 1.2%, and 1.0%, respectively. High uncontrolled eating and emotional eating scores of the 18-item Three-Factor Eating Questionnaire (TFEQ R-18) (*p* < 0.001), a high Kessler Psychological Distress Scale score (*p* < 0.001), frequent desire to overeat (*p* = 0.007), and frequent snacking (*p* = 0.003) were associated with the J-YFAS 2.0-diagnosed FA presence. The scores demonstrated significant correlations with the J-YFAS 2.0-diagnosed FA symptom count (*p* < 0.01). The highest attained body mass index was associated with the J-YFAS 2.0-diagnosed FA symptom count (*p* = 0.026). The TFEQ R-18 cognitive restraint score was associated with the J-YFAS 2.0-diagnosed FA presence (*p* < 0.05) and symptom count (*p* < 0.001), but not with the J-YFAS 2.0-diagnosed FA severity. Like the YFAS 2.0 in other languages, the J-YFAS 2.0 has a one-factor structure and adequate convergent validity and reliability.

## 1. Introduction

The idea of food addiction (FA) is receiving increased interest [1]. Evidence is emerging that certain types of foods (e.g., highly processed foods with high levels of refined carbohydrates and/or added fat) may be capable of triggering addictive-like eating behaviors (e.g., loss of control, withdrawal, and cravings) in some individuals, which can lead to significant impairment or distress, [2,3]. Obesity and eating disorders such as bulimia nervosa (BN), binge eating disorders (BED), along with psychiatric disorders such as depression, posttraumatic stress disorder, attention-deficit hyperactivity disorder, have been reported as potential correlates with FA [4,5,6]. Relevant pharmacological findings have been reported. Highly processed sweetened and fatty foods trigger a rewarding effect through the release of dopamine [7]. Repeated eating of hyper-palatable food down-regulates the dopaminergic response, resulting in impulsive and compulsive responses to food cues [8]. Food craving—an intense desire to eat a specific food—activates the hippocampus, insula, and caudate nucleus, similar to drug craving [9]. On the other hand, there has been a lot of debate regarding the extent to which food can be addictive in the same way as drugs. Controversies exist, for instance, as to whether FA represents a specific construct as addiction that is distinct from other eating disorders, such as BED, and whether neurobiological changes underlying FA behaviors are sufficiently ascertained in humans [10,11].

The Yale Food Addiction Scale (YFAS) is the most commonly used measure to assess FA, although FA is not included in the Diagnostic and Statistical Manual of Mental Disorders, 5th edition (DSM-5) [12] and controversy exists regarding its definition [11]. The original YFAS applies the DSM 4th edition (DSM-IV) diagnostic criteria for substance dependence to the consumption of highly palatable foods (e.g., chocolate, ice cream, and pizza) [13,14]. Later, the scale was replaced with the Yale Food Addiction Scale 2.0 (YFAS 2.0) in response to the revision of the Substance-Related and Addictive Disorders criteria in the DSM-5 [15]. The YFAS 2.0 additionally introduced the following four diagnostic criteria: craving, use despite interpersonal or social consequences, failure in role obligations, and use in physically hazardous situations. It also introduced a severity classification. The YFAS 2.0 is available not only in English but also in German, French, Italian, Spanish, and Arabic [16,17,18,19,20]. The YFAS 2.0 exhibits good internal consistency, as well as convergent, discriminant, and incremental validity [15,16,17,18,19,20]. Associations of the YFAS-diagnosed FA with obesity, eating disorders, and psychiatric disorders have been accumulated. The YFAS 2.0-defined FA prevalence is supposed to draw a J-shape curve according to body mass index (BMI): 3.3–15.8% in healthy general populations [15,16,17,18,19,20], 17.2–47.4% in obese population [16,21], and 15.0% in underweight population [21]. Women and patients with eating disorders (BN and BED) and mental disorders (depression, sleep disturbance, and general psychiatric status) were more likely to have FA diagnosed with the YFAS 2.0 [15,17,18,19].

The current study aimed to validate the Japanese version of YFAS 2.0 (J-YFAS 2.0). Scant evidence regarding FA is available in Asia. The previous version of YFAS was translated into Chinese [22,23] and Malay [24]. Using these questionnaires, researchers reported that a FA diagnosis was assigned to 6.9–9.2% of Chinese teenage students [22,23] and 10.4% of Malay obese adults [24]. FA prevalence in Japan has not been reported so far, to the best of our knowledge. The YFAS 2.0 has not yet been translated into Asian languages. Development of the J-YFAS 2.0 enables examining the FA prevalence in Japan, comparing it with other countries and regions, and exploring the mechanism of FA. Referring to previous research [15,16,17,18,19], we hypothesized that (1) the J-YFAS has a one-factorial structure for the 11 J-YFAS 2.0 diagnostic criteria (structural validity); (2) underweight, overweight, obesity, uncontrolled and emotional eating, frequent desire to overeat, frequent snacking, and mood and anxiety disorders are associated with the J-YFAS 2.0-diagnosed FA (convergent validity); (3) cognitive restraint in eating is not associated with the J-YFAS 2.0-diagnosed FA (discriminant validity); and (4) the internal consistency is good for the 11 J-YFAS 2.0 diagnostic criteria (reliability).

## 2. Subjects and Methods

### 2.1. Study Design

We employed a cross-sectional design. All data were collected from a questionnaire survey. The present study was completed in accordance with the Declaration of Helsinki and the Ethical Guidelines for Medical and Health Research Involving Human Subjects established by the Ministry of Education, Culture, Sports, Science and Technology and the Ministry of Health, Labour and Welfare, Japan. We obtained the approval by the Ethics Review Committee of Fujita Health University, Japan (HM17-110 and HM18-155). All subjects provided their informed written consent for participation in the present study. 

### 2.2. Subjects

This study was conducted with a convenience sample of undergraduate students from a private medical and health science university in Japan. The authors (A.O., M.M., and A.M.) explained the study purpose and methods to the students in the classes. Paper-based questionnaires were then distributed. Of the 759 students to whom the questionnaires were distributed, 752 (99%) were returned. Those who did not provide informed consent (*n* = 2) and who did not fully complete the J-YFAS 2.0 (*n* = 18) were excluded from the analysis. One student who replied to experience desire to overeat 50 times per week was excluded as this reply was a significant outlier. Consequently, we retained the remaining 731 students (96%) as the subjects. 

### 2.3. J-YFAS 2.0

As with the YFAS 2.0 [15], the J-YFAS 2.0 is a 35-item self-administered questionnaire (Appendix A). It assesses food consumption during the past 12 months. A Likert-scale ranging from 0 (never) through 7 (every day) is employed as a response option for each item. The items assess clinical impairment/distress and the following 11 diagnostic criteria: (1) eating larger amounts for a longer period than intended (consumed more than intended); (2) persistent desire or repeated unsuccessful attempts to quit eating (unable to cut down or stop); (3) spending considerable time or activity obtaining or eating food or recovering from eating (great deal of time spent); (4) giving up or reducing important social, occupational, or recreational activities due to eating (important activities given up); (5) continued eating despite knowledge of adverse consequences (use despite physical/emotional consequences); (6) development of tolerance (tolerance); (7) characteristic withdrawal symptoms (withdrawal); (8) continued eating despite interpersonal or social problems (use despite interpersonal/social problems); (9) failure to fulfil major role obligation at work, school, and home due to eating (failure in role obligation); (10) eating even in physically hazardous situations (use in physically hazardous situations); and (11) craving, strong desire, or urge for certain foods (craving). Each item is scored dichotomously based on the threshold determined by the YFAS 2.0 validation paper [15]. If any item that corresponds to the diagnostic criteria or clinical severity meets the clinical threshold, this criterion is endorsed. There are two scoring methods: the symptom count and the diagnostic threshold. For the symptom count scoring method, the diagnostic criteria for which the subjects meet are summed together. For the diagnostic threshold, the clinically significant impairment/distress criterion has to be met and two or more diagnostic criteria have to be met. The J-YFAS 2.0 FA diagnostic severity is classified as mild (2–3 criteria met plus impairment/distress), moderate (4–5 criteria met plus impairment/distress), and severe (6–11 criteria met plus impairment/distress).

For the development of the J-YFAS 2.0, the original English YFAS 2.0 [15] was translated into Japanese by the three Japanese authors (A.O., A.F., and H.Y.) and back-translated into English by an external professional translator who had no previous knowledge of the YFAS 2.0. Discrepancies between the back-translation and the original were resolved by consensus amongst the three Japanese authors and an American author (A.N.G.), who developed the original YFAS 2.0. We added two food examples, wagashi (Japanese traditional confectionery) and instant noodles (as salty snacks), in the introductory part, considering that food preference in Japan differs from western countries.

### 2.4. Variables for Convergent and Discriminant Validity

#### 2.4.1. Body Mass Index (BMI)

Each subject self-reported their current and highest attained BMI. The questionnaire included a table indicating BMI from the weights and heights so that the subjects could choose their BMI from the following options: <16.0, 16.0–16.9, 17.0–18.4, 18.5–22.9, 23.0–24.9, 25.0–29.9, and ≥30.0 kg/m^2^. No one chose <16.0 kg/m^2^ for their current or highest attained BMI. 

It was reported that Japanese tended to under-report their body weights and the tendency was more prominent among those with high BMI than those with low BMI [25]. We arranged the categorical response options for BMI to minimize the shame that subjects may feel for self-reporting their actual BMI.

#### 2.4.2. Three-Factor Eating Questionnaire Revised 18-Item Version (TFEQ R-18) 

The TFEQ R-18 is a self-assessment tool used to measure the following three types of eating behaviors: Cognitive restraint, uncontrolled eating, and emotional eating [26]. Cognitive restraint is a control over food intake in order to influence body weight and body shape [26]. Uncontrolled eating is a tendency to overeat food with the feeling of being out of control [27]. Emotional eating is a tendency to eat in response to negative emotions [27]. The higher the score is, the greater the levels of cognitive restraint, uncontrolled eating, and emotional eating are. We chose the corresponding items for the current study from the Japanese version of the original 51-item TFEQ [28]. 

#### 2.4.3. Desire to Overeat 

No validated questionnaire was available in Japanese to evaluate binge eating frequency. Thus, we asked the frequency of desiring to overeat with a single question, “How many times per week did you feel you wanted to eat more even after eating quite a lot of food during the last two hours?” The subjects filled in the number of the times.

#### 2.4.4. Snacking Frequency

A frequency of snacking (eating and drinking outside of breakfast, lunch, or dinner) was self-reported. No validated questionnaire was available in Japanese to evaluate the frequency of snacking. Thus, we developed a single question, “How many days per week are you snacking?” for this evaluation. The subjects chose one of the following options: none, 2–3 days, 4–5 days, and almost every day. The snack included foods and drinks that contained any calories. Zero-calorie drinks, such as coffee and tea without milk and sugar, and vitamin and mineral supplements were excluded from the snack. 

#### 2.4.5. Kessler Psychological Distress Scale (K6) 

The Japanese version of K6 was used as an indicator of mood and anxiety disorders [29]. A K6 score of 13 or greater was regarded as having such disorders.

### 2.5. Statistical Analyses 

Confirmatory factor analysis (CFA) was conducted to assess the one-factor structure for the 11 J-YFAS 2.0 diagnostic criteria. Clinically significant impairment/distress was not included in this CFA analysis. The model fit was evaluated with the root-mean-square error of approximation (RMSEA), comparative fit index (CFI), Tucker–Lewis index (TLI), and standardized root-mean-square residual (SRMR). For assessing the reliability, internal consistency was calculated for the 11 J-YFAS 2.0 diagnostic criteria with Kuder–Richardson’s α (KR-20). Convergent and discriminant validity was examined with chi-square test, *t*-test, analysis of variance (ANOVA), and Spearman’s rank correlation. We examined whether the current and highest attained BMI, TFEQ R-18 cognitive restraint, uncontrolled eating, and emotional eating scores, frequency of desire to overeat, snacking frequency, and K6 score were associated with the J-YFAS 2.0-diagnosed FA. Not only the presence and severity (mild, moderate, and severe) but also the symptom count was used as the J-YFAS 2.0-diagnosed FA index, given the small numbers of subjects diagnosed as having FA. We could not apply the chi-square test to examine the associations of BMI, high K6 score, and the snacking frequency with the J-YFAS 2.0-diagnosed FA severity, since more than 20% of all cells had an expected frequency of less than five. Effect size indices were calculated [30,31,32]. Subjects with missing responses were excluded from the corresponding analyses. SPSS version 23.0 (IBM, Armonk, NY, USA) and Amos Version 23.0 (IBM, Chicago, IL, USA) were used for statistical calculations.

## 3. Results 

### 3.1. Subjects’ Characteristics

Most subjects were women (78.5%, *n* = 574) (Table 1). The mean (standard deviation) age was 20.8 (1.8) years. The years and majors included fourth-year medical technology students, first- to fourth-year nursing students, and third-year medical students. Around 80% of the subjects reported normal-weight BMI, 18.5–24.9 kg/m^2^. 

### 3.2. CFA and Internal Consistency 

The RMSEA, CFI, TLI, and SRMR were 0.065, 0.904, 0.880, and 0.048, respectively. One diagnostic criterion (failure in role obligation) indicated a factor loading of 0.31 (Table 2). The other diagnostic criteria had factor loadings of 0.41 or higher. The KR-20 was 0.78 for the 11 diagnostic criteria. 

### 3.3. J-YFAS 2.0-Diagnosed FA Prevalence 

The mean J-YFAS 2.0-diagnosed FA symptom count was 0.84 (SD = 1.61; range = 0–11). The proportions of the subjects who met the threshold for each diagnostic criterion ranged from 2.9–17.0% (Table 2). A total of 24 (3.3%) subjects were regarded as having FA: 8 (1.1%) received a mild, 9 (1.2%) a moderate, and 7 (1.0%) a severe FA diagnosis using the J-YFAS 2.0 (Table 1). All subjects who were diagnosed as FA with the J-YFAS 2.0 were women. Sex was significantly associated with the J-YFAS 2.0-diagnosed FA (*p* = 0.004, Fisher’s Exact Test). 

### 3.4. Convergent and Discriminant Validity

For convergent validity, neither the current nor the highest attained BMI was associated with the presence of J-YFAS 2.0-diagnosed FA (Table 3). The highest attained BMI was associated with the J-YFAS 2.0-diagnosed FA symptom count, while the current BMI was not (Table 4). The effect size was small for the association between the highest attained BMI and the J-YFAS 2.0-diagnosed FA symptom count—the η^2^ was 0.02. A high K6 score and snacking frequency were significantly associated with the J-YFAS 2.0-diagnosed FA presence and the J-YFAS 2.0-diagnosed FA symptom count (Table 3 and Table 4, respectively). TFEQ R-18 uncontrolled eating and emotional eating scores and desire to overeat were significantly associated with the J-YFAS 2.0-diagnosed FA presence, severity, and symptom count (Table 5, Table 6 and Table 7, respectively).

For discriminant validity, there was a significant association between the TFEQ R-18 cognitive restraint score and the J-YFAS 2.0-diagnosed FA presence (Table 5). Its effect size was small—the Cohen’s d was 0.44. There was no significant association between the cognitive restraint score and the J-YFAS 2.0-diagnosed FA severity (Table 6). We found a significant correlation between the cognitive restraint score and the J-YFAS 2.0-diagnosed FA symptom count (Table 7). Its Spearman’s rank correlation coefficient was 0.143.

## 4. Discussion

We examined the J-YFAS 2.0’s properties in a sample of healthy undergraduate students in Japan. The J-YFAS 2.0 had a one-factor structure and adequate convergent validity and reliability, like the YFAS 2.0 in other languages [15,16,17,18,19,20], whereas our results were not the same as hypothesized with regard to the associations of BMI and cognitive restraint in eating with the J-YFAS 2.0-diagnosed FA. The J-YFAS 2.0-diagnosed FA prevalence was 3.3% in our subjects. Similar findings were reported from Italian and Spanish young healthy samples [18,19].

A one-factor structure was confirmed for the J-YFAS 2.0, which is the same as for the English, German, French, Italian, and Spanish YFAS 2.0 [15,16,17,18,19]. Our findings did not strictly meet the Hu and Bentler criteria, i.e., RMSEA ≤ 0.06, CFI ≥ 0.95, TLI ≥ 0.95, and SRMR ≤ 0.08 [33]. However, one or more of the four indices do not often meet the criteria [34]. The French version of the YFAS 2.0 showed a CFI of 0.887 and RMSEA of 0.083 [17]. There is the criticism that the Hu and Bentler criteria may be too stringent [35]. Our fit indices did not deviate substantially from the Hu and Bentler criteria. We thus retained the one-factor structure of the J-YFAS 2.0. Regarding the reliability of the J-YFAS 2.0, KR-20 was 0.78. This suggests the acceptable internal consistency of the J-YFAS 2.0. 

The J-YFAS 2.0-diagnosed FA prevalence was 3.3% in this study. A similar prevalence was observed in other developed countries: Italy (3.4%) and Spain (3.3%) [18,19]. The subjects’ characteristics of these three studies bear some resemblance, which might account for the similar prevalence. They were mainly young (aged about 20) and normal-weight people. Like our study, the Italian study collected the subjects from a medical school. About 80% of the subjects were female in both the Spanish and our sample. On the other hand, a web-based survey found a much higher YFAS 2.0-diagnosed FA prevalence, 9.7%, among German-speaking university students with the similar age and BMI [16]. This could imply that not only biological characteristics but also cultural differences are associated with YFAS 2.0-diagnosed FA, although it is possible that the web-based survey received considerable attention from those with FA and obtained their participation. Similar to previous reports in the U.S. [15] and Italy [18], women exhibited a significantly greater YFAS 2.0-diagnosed FA prevalence than men in our study. This suggests a sex difference in YFAS 2.0-diagnosed FA, which should be further investigated in future studies.

The YFAS 2.0-diagnosed FA prevalence was high in overweight, obese, and underweight people in the U.S., Germany, France, Italy, Spain, and Egypt [15,16,17,18,19,20,21]. Contrary to these findings, both the current and highest attained BMI did not demonstrate an explicit association with the J-YFAS 2.0-diagnosed FA in the present study. We only found that the subjects with the highest attained BMI of 25.0–29.9 had a greater J-YFAS 2.0-diagnosed FA symptom count than those with the highest attained BMI of 17.0–24.9. However, its effect size was small—the η^2^ was only 0.02. One possible reason for the finding might be the small numbers of our subjects with overweight, obesity, and extreme underweight. Current overweight and obesity were declared only by nearly 4% of the subjects. This reflected the fact that Japan has a much lower prevalence of overweight and obesity than other countries where the YFAS 2.0 has been validated [36]. Perhaps, some subjects in our study could have under-reported their BMI [37], although we arranged the categorical response options for BMI to minimize the shame that subjects may feel for self-reporting their actual BMI. Consequently, the low prevalence of overweight and obesity exerted a floor effect, diminishing the association between BMI, especially overweight and obesity, and the J-YFAS 2.0-diagnosed FA. Another possible reason is that the causes to affect BMI are multifactorial and different by region. We did not examine all causes that potentially affected BMI more strongly than FA. For instance, some researchers pointed out that social norms (pressure) might drive the young Japanese women’s desire for slimming [38,39,40]. They could have more impact on BMI than FA among our subjects. Our subjects involved medical, nursing, and medical technology students. They must have a greater knowledge of health, nutrition, and exercise than the normal population, which may have skewed the association between BMI, especially the current BMI, and the J-YFAS 2.0-diagnosed FA. Development of the J-YFAS 2.0 improves the examination of FA in Japan where the prevalence of obesity is much lower than the western countries [36]. This may help elucidate our understanding of the impact of FA on body weight. Although FA was initially applied to understanding obesity, controversy remains over how much FA explains obesity [10,11].

Other variables hypothetically related to the convergent validity of the J-YFAS 2.0 showed significant associations with the J-YFAS 2.0-diagnosed FA as we expected. The TFEQ R-18 uncontrolled eating and emotional eating scores and desire to overeat were associated with the J-YFAS 2.0-diagnosed FA presence, severity, and symptom count in our study, as hypothesized based on the previous studies [15,16,17,18,19]. One study limitation is that we could not assess binge eating itself which was positively and moderately associated with the YFAS-diagnosed FA [41]. However, our findings regarding the desire to overeat and snacking would suggest the relationship between compulsive eating and FA. A desire to overeat forms a part of binge eating. Highly processed sweetened foods, which has been reported to be potentially related to FA [42,43,44], are often chosen for snacking in Japan [45]. We found that a high K6 score, which implied mood and anxiety disorders, is associated with the presence and higher symptom count of the J-YFAS 2.0-diagnosed FA. Some previous studies showed associations of psychopathological disorders [19] and depressive symptoms [18] with the YFAS 2.0-diagnosed FA. A recent systematic review suggested a positive, moderate association of the YFAS-diagnosed FA with depression and anxiety [41]. Our finding was consistent with them.

Regarding the discriminant validity, we hypothesized that cognitive restraint in eating was a different entity from FA, referring to the idea that the YFAS 2.0 does not simply measure an intention and a failure to restrict food consumption [15,16]. Our findings exhibited an inconsistency in the association of cognitive restraint in eating with the FA presence, severity, and symptom count. In our sample, the association between cognitive restraint and the J-YFAS 2.0-diagnosed FA would not be so strong even if the association existed. Previous findings are also inconsistent regarding the association. It was reported in France and Italy that the YFAS 2.0-diagnosed FA was associated with a high level of cognitive restraint [17,18]. The Italian researchers mentioned the possibility that addictive-like eating and restricting food consumption could coexist in patients with anorexia nervosa [18]. We could not ascertain this possibility in our study since we did not examine whether the subjects suffered from anorexia. Further research would be necessary to examine the role of anorexia in the association between cognitive restraint and FA.

There are several limitations to the interpretation of our findings. First, the current study employed a convenience sample that was dominated by young, under- and normal-weight, female, healthy undergraduate students. For a generalization of the present findings, the J-YFAS 2.0 should be tested for different-age groups, obese individuals, and patients with eating disorders. Second, we were not able to include all kinds of validity and reliability. For instance, we did not address incremental validity and test-retest reliability. Third, we used our original questions to assess the desire to overeat and frequency of snacking. For instance, the Binge Eating Scale (BES) [46] and the Eating Behavior Patterns Questionnaire (EBPQ) [47] are the validated tools to evaluate binge eating and snacking, respectively. We did not use them since they were not translated into and validated in Japanese. This may limit the comparison of our results with the others. Finally, as FA has not yet been recognized in the DSM-5, we could not define the standard of psychiatrist-diagnosed FA.

As mentioned in the introduction, the conceptual construct of FA and the neurobiological changes underpinning it remain controversial [10,11]. Development of the J-YFAS 2.0 would facilitate research on FA in Japan where prevalence of overweight and obesity is much lower than the western countries [36]. This would contribute to specifying the conceptual construct of FA and the neurobiological changes related to FA.

## 5. Conclusions

The J-YFAS 2.0 had a one-factor structure and adequate convergent validity and reliability, like the YFAS 2.0 in other languages. Further studies are necessary to confirm the discriminant validity of the J-YFAS 2.0.

## Figures and Tables

**Table 1 nutrients-11-00687-t001:** Subject characteristics (*n* = 731).

Characteristics	Frequency (%) or Mean (SD)
Sex	
Men	156 (21.3%)
Women	574 (78.5%)
Age (year)	20.8 (1.8)
Years and Majors	
Fourth-year medical technology students	149 (20.4%)
First-year nursing students	142 (19.4%)
Second-year nursing students	132 (18.1%)
Third-year medical students	111 (15.2%)
Fourth-year nursing students	99 (13.5%)
Third-year nursing students	98 (13.4%)
Current body mass index (BMI) (kg/m^2^)	
16.0–16.9	17 (2.3%)
17.0–18.4	108 (14.8%)
18.5–22.9	521 (71.3%)
23.0–24.9	57 (7.8%)
25.0–29.9	21 (2.9%)
30 and above	6 (0.8%)
Highest attained BMI (kg/m^2^) *	
16.0–16.9	3 (0.4%)
17.0–18.4	60 (8.2%)
18.5–22.9	493 (67.4%)
23.0–24.9	117 (16.0%)
25.0–29.9	51 (7.0%)
30 and above	7 (1.0%)
Kessler Psychological Distress Scale (K6) score	4.6 (4.5)
13 or greater	45 (6.2%)
Three-factor Eating Questionnaire-R 18 (TFEQ R-18) score	
Cognitive restraint	37.0 (20.2)
Uncontrolled eating	35.5 (19.9)
Emotional eating	29.7 (27.5)
Desire to overeat	0.5 (1.0) (Range: 0–7)
Snacking frequency per week	
None	89 (12.2%)
2–3 days	276 (37.8%)
4–5 days	157 (21.5%)
Almost every day	208 (28.5%)
J-YFAS 2.0-diagnosed food addiction (FA)	
No FA	707 (96.7%)
Mild FA	8 (1.1%)
Moderate FA	9 (1.2%)
Severe FA	7 (1.0%)

SD: standard deviation. There were missing responses for sex (*n* = 1), age (*n* = 1), current BMI (*n* = 1), K6 (*n* = 4), the TFEQ R-18 cognitive restraint (*n* = 8), uncontrolled eating (*n* = 12), and emotional eating (*n* = 2), desire to overeat (*n* = 1), and snacking frequency (*n* = 1). * Highest attained BMI means the highest weight ever (when not pregnant) during the lifetime.

**Table 2 nutrients-11-00687-t002:** Diagnostic criteria of the Japanese version of the Yale Food Addiction Scale 2.0 (*n* = 731).

Diagnostic Criteria	Met Criteria	Did Not Meet Criteria	Factor Loading
Consumed more than intended	82 (11.2%)	649 (88.8%)	0.57 ***
Unable to cut down or stop	124 (17.0%)	607 (83.0%)	0.52 ***
Great deal of time spent	30 (4.1%)	701 (95.9%)	0.45 ***
Important activities given up	25 (3.4%)	706 (96.6%)	0.41 ***
Use despite physical/emotional consequences	45 (6.2%)	686 (93.8%)	0.55 ***
Tolerance	31 (4.2%)	700 (95.8%)	0.50 ***
Withdrawal	90 (12.3%)	641 (87.7%)	0.62 ***
Use despite interpersonal/social problems	98 (13.4%)	633 (86.6%)	0.54 ***
Failure in role obligation	29 (4.0%)	702 (96.0%)	0.31 ***
Use in physically hazardous situations	42 (5.7%)	689 (94.3%)	0.56 ***
Craving	21 (2.9%)	710 (97.1%)	0.50 ***
Impairment/distress	29 (4.0%)	702 (96.0%)	

*** *p* < 0.001, calculated with confirmatory factor analysis.

**Table 3 nutrients-11-00687-t003:** Associations of body mass index (BMI), the Kessler Psychological Distress Scale (K6) score, and snacking frequency with the J-YFAS 2.0-diagnosed food addiction (FA) absence/presence.

	FA Absent (*n* = 707)	FA Present (*n* = 24)	Chi-Square	*p* Value	Effect Size (*V*)
**Current BMI (kg/m^2^)**					
16.0–16.9	16 (94.1%)	1 (5.9%)	1.421	0.922	0.04
17.0–18.4	105 (97.2%)	3 (2.8%)			
18.5–22.9	503 (96.5%)	18 (3.5%)			
23.0–24.9	55 (96.5%)	2 (3.5%)			
25.0–29.9	21 (100%)	0 (0%)			
30 and above	6 (100%)	0 (0%)			
**Highest attained BMI (kg/m^2^) ***					
16.0–16.9	3 (100%)	0 (0%)	1.522	0.911	0.05
17.0–18.4	58 (96.7%)	2 (3.3%)			
18.5–22.9	478 (97.0%)	15 (3.0%)			
23.0–24.9	113 (96.6%)	4 (3.4%)			
25.0–29.9	48 (94.1%)	3 (5.9%)			
30 and above	7 (100%)	0 (0%)			
**K6 score**					
12 or less	665 (97.5%)	17 (2.5%)	22.565	<0.001	0.18
13 or greater	38 (84.4%)	7 (15.6%)			
**Snacking Frequency Per Week**					
None	89 (100%)	0 (0%)	13.855	0.003	0.14
2–3 days	272 (98.6%)	4 (1.4%)			
4–5 days	151 (96.2%)	6 (3.8%)			
Almost every day	194 (93.3%)	14 (6.7%)			

Chi-square test was used. The numbers of missing responses were as follows: current BMI (*n* = 1), K6 score (*n* = 4), and snacking frequency (*n* = 1). * Highest attained BMI means the highest weight ever (when not pregnant) during the lifetime.

**Table 4 nutrients-11-00687-t004:** Associations of body mass index (BMI), the Kessler Psychological Distress Scale (K6) score, and snacking frequency with the J-YFAS 2.0-diagnosed food addiction (FA) symptom count (*n* = 731).

	FA Symptom Count	F/*t* Value	*p* Value	Pairwise Difference ^a^	Effect Size (η^2^)/(d)
**Current BMI (kg/m^2^)**					
16.0–16.9	1.0 (2.7)	1.375	0.231		0.01
17.0–18.4	0.6 (1.2)				
18.5–22.9	0.8 (1.6)				
23.0–24.9	1.1 (1.8)				
25.0–29.9	1.4 (2.1)				
30 and above	1.3 (1.5)				
**Highest attained BMI (kg/m^2^) ***					
16.0–16.9 (1)	0.7 (0.6)	2.555	0.026	(2), (3), (4) < (5)	0.02
17.0–18.4 (2)	0.6 (1.7)				
18.5–22.9 (3)	0.8 (1.6)				
23.0–24.9 (4)	0.7 (1.1)				
25.0–29.9 (5)	1.5 (2.3)				
30 and above (6)	1.1 (1.5)				
**K6 score**					
12 or less	0.8 (1.4)	−3.060	0.004		0.95
13 or greater	2.2 (3.2)				
**Snacking frequency per week**					
None (1)	0.4 (0.7)	15.986	<0.001	(1), (2) < (3), (4)	0.06
2–3 days (2)	0.5 (1.0)				
4–5 days (3)	1.0 (1.9)				
Almost every day (4)	1.4 (2.1)				

Analysis of variance (for BMI and snacking frequency) and *t*-test (for K6 score) were used. FA symptom counts are shown as mean (standard deviation). The numbers of missing responses were as follows: Current BMI (*n* = 1), K6 score (*n* = 4), and snacking frequency (*n* = 1). ^a^ Pairwise differences were of *p* < 0.05 (Bonferroni corrected). * Highest attained BMI means the highest weight ever (when not pregnant) during the lifetime.

**Table 5 nutrients-11-00687-t005:** Associations of the 18-item Three-Factor Eating Questionnaire (TFEQ R-18) scores and frequency of desiring to overeat with the J-YFAS 2.0-diagnosed food addiction (FA) absence/presence.

	FA Absent (*n* = 707)	FA Present (*n* = 24)	*t* Value	*p* Value	Effect Size (d)
Cognitive restraint	36.7 (20.1)	45.7 (21.7)	−2.097	0.036	0.44
Uncontrolled eating	34.5 (19.1)	64.1 (23.1)	−7.246	<0.001	1.54
Emotional eating	28.6 (26.4)	63.9 (34.6)	−4.959	<0.001	1.32
Desire to overeat	0.41 (0.91)	1.7 (2.0)	−2.965	0.007	1.28

*t*-test was used. TFEQ R-18 scores and frequency of desiring to overeat are shown as mean (standard deviation). The numbers of missing responses were as follows: TFEQ R-18 cognitive restraint, *n* = 8 (7 from FA absent, 1 from FA present); uncontrolled eating, *n* = 12 (11 from FA absent, 1 from FA present); emotional eating, *n* = 2 (all from FA absent); and desire to overeat, *n* = 1 (from FA absent).

**Table 6 nutrients-11-00687-t006:** Associations of the 18-item Three-Factor Eating Questionnaire (TFEQ R-18) scores and frequency of desiring to overeat with the J-YFAS 2.0-diagnosed food addiction (FA) severity.

	FA Absent (*n* = 707)	Mild FA (*n* = 8)	Moderate FA (*n* = 9)	Severe FA (*n* = 7)	F Value	*p* Value	Pairwise Difference ^a^	Effect Size (η^2^)
Cognitive restraint	36.7 (20.1)	43.1 (17.8)	48.6 (24.3)	45.2 (25.6)	1.56	0.197		0.01
Uncontrolled eating	34.5 (19.1)	56.0 (24.5)	62.5 (20.5)	75.1 (23.2)	18.80	<0.001	1 < 2,3,4	0.07
Emotional eating	28.6 (26.4)	43.1 (37.3)	74.1 (26.1)	74.6 (34.4)	16.06	<0.001	1 < 3,4	0.06
Desire to overeat	0.4 (0.9)	0.6 (1.2)	1.8 (1.5)	2.7 (2.9)	19.53	<0.001	1 < 3, 4; 2 < 4	0.07

Analysis of variance was used. TFEQ R-18 scores and frequency of desiring to overeat are shown as mean (standard deviation). The numbers of missing responses were as follows: TFEQ R-18 cognitive restraint, *n* = 8 (7 from FA absent, 1 from Moderate FA); uncontrolled eating, *n* = 12 (11 from FA absent, 1 from Moderate FA); emotional eating, *n* = 2 (all from FA absent); and desire to overeat, *n* = 1 (from FA absent). ^a^ Pairwise differences were of *p* < 0.05 (Bonferroni corrected). 1 = No FA, 2 = Mild FA, 3 = Moderate FA, 4 = Severe FA.

**Table 7 nutrients-11-00687-t007:** Spearman’s rank correlation coefficients among the J-YFAS 2.0-diagnosed food addiction (FA) symptom count, the 18-item Three-Factor Eating Questionnaire (TFEQ R-18) scores, and frequency of desiring to overeat (*n* = 731).

	FA Symptom Count	Cognitive Restraint	Uncontrolled Eating	Emotional Eating	Desire to Overeat
FA symptom count					
Cognitive restraint	0.143 ***				
Uncontrolled eating	0.403 ***	0.248 ***			
Emotional eating	0.296 ***	0.258 ***	0.619 ***		
Desire to overeat	0.277 ***	0.039	0.449 ***	0.361 ***	

*** *p* < 0.001. The numbers of missing responses were as follows: TFEQ R-18 cognitive restraint (*n* = 8), uncontrolled eating (*n* = 12), emotional eating (*n* = 2), and desire to overeat, (*n* = 1).

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
