# Peer review of "Validation of the Japanese Version of the Yale Food Addiction Scale 2.0 (J-YFAS 2.0)"

_nutrients, 2019, doi:10.3390/nu11030687_

Round 1

Reviewer 1 Report

The goal of this study was to assess the validity of the Japanese version of the YFAS 2.0.  The authors found that in a sample of mostly normal-weight, mostly female Japanese college students, YFAS 2.0 had a one-factor structure, and had adequate convergent validity and reliability. Discriminant validity was not confirmed using the construct of cognitive restraint.

This manuscript is well written and executed. Only minor comments are included below.

Introduction

In the first paragraph, it would be helpful to add some specific examples of adverse health and psychological correlates associated with FA (e.g., Meule and Gearheardt, 2014) to further highlight the importance of that construct.

Methods

Ln 120: I am curious as to why the authors chose to use self-reported BMI, instead of self-reported height and weight, for the highest attained and current BMI?

Author Response

Response to Reviewer 1 Comments

Comments

The goal of this study was to assess the validity of the Japanese version of the YFAS 2.0.  The authors found that in a sample of mostly normal-weight, mostly female Japanese college students, YFAS 2.0 had a one-factor structure, and had adequate convergent validity and reliability. Discriminant validity was not confirmed using the construct of cognitive restraint.

This manuscript is well written and executed. Only minor comments are included below.

We appreciate you for highly evaluating our manuscript.

Point 1: Introduction

In the first paragraph, it would be helpful to add some specific examples of adverse health and psychological correlates associated with FA (e.g., Meule and Gearheardt, 2014) to further highlight the importance of that construct.

Response 1: We specified the adverse health and psychological correlates associated with FA in the first and second paragraph.

(Lines 40-42) “Obesity, eating disorders (bulimia nervosa (BN), binge eating disorders (BED), etc.), and psychiatric disorders (depression, posttraumatic stress disorder, attention-deficit hyperactivity disorder, etc.) have been reported as potential correlates with FA [4-6].”

(Lines 63-64) “Associations of the YFAS-diagnosed FA with obesity, eating disorders, and psychiatric disorders have been accumulated.”

(Lines 67-69) “Women and patients with eating disorders (BN and BED) and mental disorders (depression, sleep disturbance, and general psychiatric status) were more likely to have FA diagnosed with the YFAS 2.0 [15,17-19].”

Point 2: Methods

Ln 120: I am curious as to why the authors chose to use self-reported BMI, instead of self-reported height and weight, for the highest attained and current BMI?

Response 2: We introduced a categorical self-report of BMI for the sake of the potential under-report of the weight and the shame the subjects would feel when self-reporting the actual numbers of body mass. To clarify this, we made the following revisions.

(Subjects and Methods, lines 134-136) “The questionnaire included a table indicating BMI from the weights and heights so that the subjects could choose their BMI from the following options: <16.0, 16.0–16.9, 17.0–18.4, 18.5–22.9, 23.0–24.9, 25.0–29.9, and ≥30.0 kg/m2.”

(Subjects and Methods, lines 138-141) “It was reported that Japanese tended to under-report their body weights and the tendency was more prominent among those with high BMI than those with low BMI [25]. We arranged the categorical response options for BMI for the sake of the shame the subjects would feel for self-reporting their actual BMI.”

(Discussion, lines 286-288) “Perhaps, some subjects in our study could have under-reported their BMI [37], although we arranged the categorical response options for BMI for the sake of the shame the subjects would feel for self-reporting their actual BMI.”

Reviewer 2 Report

The paper presents the validation of the Japanese version of the Yale Food Addiction Scale (2.0). It is well presented and provides a useful method to assess YFAS-diagnosed ‘food addiction’ within Japanese samples. My main concerns centre around the assumption that the YFAS provides a valid tool for the assessment of food addiction; this has been widely debated throughout the scientific literature and so I feel that the paper could be improved by providing a brief discussion of these controversies.

Specifically, I recommend the following changes:

·        Lines 34 – 38 the authors state that evidence is emerging that certain types of foods may trigger addictive-like eating. They support this by referring to similarities in brain activation caused by drugs and food. However, there is a lot of debate regarding the extent to which food can be ‘addictive’ in the same way as drugs. I would therefore encourage the authors to acknowledge some of the controversies surrounding the concept of food addiction in order to present a balanced and up-to-date reflection on current literature.

·        On line 63, the authors suggest that developing the J-YFAS 2.0 would enable the examination of the prevalence of FA in Japan. Given that a) there is no agreed-upon clinical criteria for FA and b) the extent that the DSM criteria for SRAD can be used to assess eating behaviour is limited, this statement is overstating the use of the J-YFAS 2.0 (or any version of the YFAS). I would suggest that, throughout the paper, the authors refer to ‘YFAS-diagnosed food addiction’ rather than simply ‘food addiction’.

·        Section 2.4.3 – Desire to overeat. Please state the response options (if any) that were given to participants.

·        In the Discussion section, the authors should acknowledge the limitations of the YFAS and the FA model in general (see Ziauddeen, Farooqi & Fletcher, 2012 for a review).

Author Response

Response to Reviewer 2 Comments

Comments

The paper presents the validation of the Japanese version of the Yale Food Addiction Scale (2.0). It is well presented and provides a useful method to assess YFAS-diagnosed ‘food addiction’ within Japanese samples. My main concerns centre around the assumption that the YFAS provides a valid tool for the assessment of food addiction; this has been widely debated throughout the scientific literature and so I feel that the paper could be improved by providing a brief discussion of these controversies.

We appreciate you for highly evaluating our manuscript.

Point 1: Lines 34 – 38 the authors state that evidence is emerging that certain types of foods may trigger addictive-like eating. They support this by referring to similarities in brain activation caused by drugs and food. However, there is a lot of debate regarding the extent to which food can be ‘addictive’ in the same way as drugs. I would therefore encourage the authors to acknowledge some of the controversies surrounding the concept of food addiction in order to present a balanced and up-to-date reflection on current literature.

Response 1: We introduced the above-written debate and controversy in the first paragraph of the introduction. We appreciate you for kindly introducing a review by Ziauddeen, Farooqi, & Fletcher (2012) (reference no. 10) in your review comment no. 4. In addition, we added a more recent review by Fletcher & Kenny (2018) (reference no. 11). These 2 papers support the debate and controversy.

(Lines 47-51) “On the other hand, there has been a lot of debate regarding the extent to which food can be addictive in the same way as drugs. Controversies exist as to, for instance, whether FA represents a specific construct as addiction that is distinct from other eating disorders, such as BED, and whether neurobiological changes underlying FA behaviors are sufficiently ascertained in humans [10,11].”

Point 2: On line 63, the authors suggest that developing the J-YFAS 2.0 would enable the examination of the prevalence of FA in Japan. Given that a) there is no agreed-upon clinical criteria for FA and b) the extent that the DSM criteria for SRAD can be used to assess eating behaviour is limited, this statement is overstating the use of the J-YFAS 2.0 (or any version of the YFAS). I would suggest that, throughout the paper, the authors refer to ‘YFAS-diagnosed food addiction’ rather than simply ‘food addiction’.

Response 2: We changed from “food addiction (FA)” to ‘J-YFAS 2.0-diagnosed FA” or “YFAS 2.0-diagnosed FA” or “YFAS-diagnosed FA” throughout our paper when it is necessary to differentiate general FA from FA diagnosed with the J-YFAS 2.0, or YFAS 2.0 or YFAS in other languages.

(Lines 22, 26, 27, 28, 29, 30, 63, 69, 80, 81, 121-122, 173, 174-175, 177, 186 (Table 1), 197, 198, 201, 202, 203, 206, 207, 208, 210, 212, 215, 217, 218, 221, 225, 231, 237, 244, 253, 254, 269-270, 272, 274-275, 275-276, 279-280, 281, 290, 297, 302, 303-304, 310, 312, 317).

Point 3: Section 2.4.3 – Desire to overeat. Please state the response options (if any) that were given to participants.

Response 3: We asked the subjects to fill in the actual number of given one open question ‘times per week’ as a continuous data to participants. No response options were given.

(Subjects and Methods, lines 153-154) “The subjects filled in the number of the times.”

Point 4: In the Discussion section, the authors should acknowledge the limitations of the YFAS and the FA model in general (see Ziauddeen, Farooqi & Fletcher, 2012 for a review).

Response 4: In response to your comment no. 1, we introduced the debate and controversy referring to the limitations of the YFAS and FA model. This is also the response to this review comment. In this line, we add discussion on the prospects which development of the J-YFAS 2.0 produce to shed light on the debate and controversy.

There is also a controversy how much FA can explain obesity since FA was initially applied to understanding obesity. We added discussion on this point.

(Introduction, lines 47-51) “On the other hand, there has been a lot of debate regarding the extent to which food can be addictive in the same way as drugs. Controversies exist as to, for instance, whether FA represents a specific construct as addiction that is distinct from other eating disorders, such as BED, and whether neurobiological changes underlying FA behaviors are sufficiently ascertained in humans [10,11].”

(Discussion, lines 297-300) “Development of the J-YFAS 2.0 enables examining FA in Japan where prevalence of obesity is much lower than the western countries [36]. This may contribute to solving the impact of FA on body weight. Although FA was initially applied to understanding obesity, there is still a controversy how much FA explains obesity [10,11].”

(Discussion, lines 332-336) “As mentioned in the introduction, it is still controversial regarding the conceptual construct of FA and neurobiological changes related to FA [10,11]. Development of the J-YFAS 2.0 would facilitate research on FA in Japan where prevalence of overweight and obese is much lower than the western countries [36]. This would contribute to specifying the conceptual construct of FA and neurobiological changes related to FA.”
